# Echoes of the Heart: Henry James's Evocation of Edgar Allan Poe in "The Aspern Papers"

## Leonardo Buonomo

Department of Humanities, University of Trieste, 34123 Trieste, Italy; buonomo@units.it

**Abstract:** This essay re-examines Henry James's complex relationship with Edgar Allan Poe by focusing on the echoes of one of Poe's most celebrated tales, "The Tell-Tale Heart" (1843), that later reverberate in James's "The Aspern Papers" (1888). It highlights the similarities, both in mindset and behavior, between the two stories' devious and deranged first-person narrators, whose actions result in the death of a fellow human being. It further discusses the narrators' fear and refusal of their own mortality, which finds expression in their hostility, and barely contained revulsion against a man (in "The Tell-Tale Heart") and a woman (in "The Aspern Papers"), whose principal defining traits are old age and physical decay.

**Keywords:** Henry James; Edgar Allan Poe; "The Aspern Papers"; "The Tell-Tale Heart"; narrative voice; old age; violation of privacy; visual scrutiny

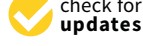



Revisiting his childhood in *A Small Boy and Others* (1913), Henry James paid belated homage to Edgar Allan Poe, an author he had contemptuously dismissed many years earlier, at the outset of his literary career. As is well known, in his 1876 essay on Charles Baudelaire, an admirer and translator of Poe, the young Henry James had killed two birds with one stone, so to speak, by stating that "[an] enthusiasm for Poe is the mark of a decidedly primitive stage of reflection" (James 1904, p. 60). Ironically, the very harshness of James's early criticism (another example is his notorious pan of Walt Whitman's *Drum-Taps*), aligns him with Poe, himself often a brutal reviewer. Moreover, only three years after the appearance of the Baudelaire essay, James had specifically acknowledged Poe's critical insight, however frequently marred, in his view, by pedantry or vulgar animosity, in his book-length study *Hawthorne* (1879). Calling him "a man of genius", whose "intelligence was frequently great", James had singled out for praise Poe's early perceptive assessment of Hawthorne's worth and had concurred in his reservations about Hawthorne's fondness for allegory (James 1956, pp. 50–51). As Adeline Tintner (1976), among others, has demonstrated, James occasionally borrowed from Poe, most conspicuously perhaps in the novella "Glasses" (1896), based at least in part on Poe's tale "The Spectacles" (1844).[1] And in what was, arguably, the crowning achievement of his career as a novelist—*The Golden Bowl* (1904)—James attributed an appreciation of Poe (specifically, of *The Narrative of Arthur Gordon Pym*, 1838) to the finely tuned and highly sophisticated sensitivity of Prince Amerigo.[2]

In *A Small Boy*, from the perspective of old age, James recalled with genuine fondness and deeply-felt gratitude the sheer delight he had experienced as a child while reading—or listening to his brother William's reading of—Poe's tales and poems, savoring the sound of every word. Indeed, as evoked in *A Small Boy*, the presence of Poe in Henry's life was more pervasive, and more intensely felt, than that of some authors he had actually met in the flesh. An "enthusiasm for Poe" had been passed from brother to brother, soon becoming a bond between them, based on a shared aesthetic, emotional, and performative experience, with the siblings taking turns mounting on "little platforms" to declaim such Poe classics as "The Raven" and "Annabel Lee" (James 2016, p. 40). In a sense, the brothers had played at *being* Poe, trying to conjure up in their own voices the auditory quality Poe

had so memorably captured in his writing. Like so many of Poe's male heroes, Henry and William had been haunted, as it were: "he lay upon our tables and resounded in our mouths, while we communed to satiety, even for boyish appetites, over the thrill of his choicest pages" (James 2016, p. 40). Thinking back on those days, Henry James could only mention, almost incredulously, "the legend of the native neglect" (James 2016, p. 40) of Poe, if the James household and its circle were any measures of Poe's popularity in the United States. In effect, in this section of his autobiography, James simultaneously acknowledged Poe as a powerful influence on his developing imagination and reclaimed him as a national treasure. Poe's influence can indeed be felt in several of James's stories and novels, as well as in his theories of fiction. A great deal has been written on the intertextual dialogue between Poe and James over the years, but not, specifically, on the links between the two stories—Poe's "The Tell-Tale Heart" and James's "The Aspen Papers"—whose similarities in tone and themes I will discuss in this article. In order to situate my reading, I begin by offering an overview of critical opinions on the relationship between Poe and James over several decades.

Numerous claims have been made over the years, sometimes positing direct lines of influence between Poe and James, sometimes pointing to correspondences or proximities in subject matter, characterization, setting, or conception between their works. Allen Tate noted that Poe's "insistence upon unity of effect, from first word to last, in the famous review of Hawthorne's *Twice-Told Tales*" anticipated "the high claims of James in his essay *The Art of Fiction*" (Tate 1950, p. 1). Robert Regan called attention to the atmospheric resemblance between James's ghost stories, such as "The Turn of the Screw" and "The Jolly Corner", and Poe's tales of terror, and argued that in James's works, "just as in 'William Wilson' and 'The Fall of the House of Usher', we confront ghostly figures who are the objectified, projected compulsions of the 'real' characters" (Regan 1967, p. 9). Joel Salzberg found a Gothic kinship between the male protagonists of Poe's "Ligeia" and James's "The Beast in the Jungle" (Salzberg 1972, pp. 108–14). Burton Pollin, in what is still one of the most comprehensive studies of the Poe–James relationship, identified allusions to "The Raven", "The Masque of the Red Death", and the poem "The Haunted Palace" in *The Sacred Fount* (1901), as well as a resemblance between the mentally unstable narrator of that novel and his counterpart in "The Fall of the House of Usher". Pollin argued that these traces and, even more significantly, the reference to Poe in *The Golden Bowl* (originally called "Mystification", like one of Poe's tales), were evidence of James's re-reading and re-evaluation of the author he had so publicly dismissed many years earlier. Indeed, according to Pollin, the very words James finally chose as the definitive title of the novel, and its central symbol, may have been suggested to him by "Lenore", one of the poems by Poe that, as evidenced by his fond recollection in *A Small Boy and Others*, he had never forgotten. Finally, Pollin contended that Poe's treatment of the double motif in "William Wilson" was one of the sources of inspiration for James's "The Jolly Corner" (Pollin 1973, pp. 234–37, 241–42).

In 1980, Elsa Nettels (1980) highlighted the points of contact between Poe's and James's (1909) theory of fiction, as did Agostino Lombardo (1990) ten years later. Christopher Brown focused on "The Masque of the Red Death" as a possible source for *The Portrait of a Lady*, noting how both Prince Prospero's castellated abbey and Osmond's Palazzo Roccanera, equally shaped by the taste and personality of their masters, turn out to be spaces of confinement and, ultimately, entombment (Brown 1981, p. 7). Howard Kerr, expanding on brief suggestions previously made by Cornelia P. Kelley and Leon Edel, made a case

---

1   On the similarities between "Glasses" and "The Spectacles", see also Donatella Izzo's *Portraying the Lady* (Izzo 2001, pp. 123–24) and my own "A Woman's Face" (Buonomo 2003, pp. 51–52). In addition to "The Spectacles", Izzo mentions Poe's tale "The Sphinx" as a possible source for "Glasses" (Izzo 2001, p. 243). I believe Poe's "The Oval Portrait" may have been another influence (Buonomo 2003, p. 52).

2   Admittedly, only a few years later, in the preface to *The Altar of the Dead* (volume 17 of the New York Edition, 1909), James described the climax of *Arthur Gordon Pym* as a failure (James 1934, pp. 256–57), but his critique is nonetheless evidence of the continuing presence of Poe as a term of comparison in his reflections about the art of writing. That presence is all the more remarkable if one considers the neglect into which Poe had fallen in the United States at the time.

for the influence of "The Fall of the House of Usher" on James's "The Ghostly Rental" (Kerr 1983, pp. 140–45). Adeline Tintner, in a brief aside, observed that "Usher" is "behind 'The Jolly Corner'" (Tintner 1989, p. 188).

More recently, Peter Rawlings has drawn attention to the similarity between Poe's "paradigmatic American statement on the relation between the picturesque and the grotesque" and "the projects of James's travel writing" (Rawlings 2004, p. 176). Stacey Margolis has offered an insightful comparative analysis of Poe's "The Man That Was Used Up" and James's "The Papers", showing how both authors were concerned with, and appalled by, the ways in which public opinion can create and consume celebrity (Margolis 2009). Mark McGurl, in addition to renewing interest in the reference to *Arthur Gordon Pym* in *The Golden Bowl*, has intriguingly suggested that the golden bowl recalls the purloined letter in Edgar Allan Poe's eponymous tale, which "like the bowl is evidence of adultery, and . . . lends its title to the text in which it appears" (McGurl 2001, p. 32).[3]

A possible link between Poe's work and James's "The Aspern Papers", in particular, has been discussed by Gerald Kennedy and Leland Person. As announced by the subtitle of his essay ("A Speculation"), Kennedy conjectures that the likeliest American prototype of the elusive figure of Jeffrey Aspern, a romantic poet of note in the early decades of the nineteenth century, was none other than Poe.[4] According to Kennedy, the poor opinion of Poe that James had voiced in the essay on Baudelaire resonates, unchanged, in "The Aspern Papers". In Kennedy's view, James's damning portrayal of the story's narrator suggests that he is probably as bad a judge of literary quality as he is of human nature: "Rather than a 'distinguished presence' emerging from an aesthetic vacuum, perhaps the idolized Aspern is no more than a vacuous poet—a poet, James may have thought, very much like Edgar Allan Poe". An additional link between Poe and "The Aspern Papers" might be found in the description of the—by then, very old—Poe devotee Sarah Helen Whitman in Thomas Wentworth Higginson's *Short Studies of American Authors* (1879), a description that presented remarkable similarities to James's portrayal of Juliana Bordereau. Moreover, Higginson's book included an unfavorable evaluation of James's production that we know James read[5] and which must have stung him all the more when compared to Higginson's appreciative homage to Poe in the same volume (Kennedy 1973, pp. 17–18).

More relevant to the present essay are Leland Person's focus on the psychology of the narrator in "The Aspern Papers" and William Veeder's acknowledgment of Poe's influence on James in his essay "The Aspern Portrait". Person contends that the narrator's view of Juliana "suggests an exaggerated male fear of women, recalling Hawthorne's Veiled Lady in *The Blithedale Romance*, or one of Poe's women-returned-from-the-dead". But it is the narrator's posture and voice that most recall Poe's unstable mediators between the reader and the subject matter: "Like one of Poe's compulsive characters", Person notes, "James's protagonist seems to stand apart from himself, commenting upon actions that, even as narrator, he is powerless to prevent, his uncanny awareness of detail bespeaking the climax that approaches" (Person 1986, pp. 24–25). Finally, in his concluding remarks on "The Aspern Papers", William Veeder asks us to recognize the resemblance between James's "failed protagonists" and those of his "American precursor", namely Edgar Allan Poe, whose male heroes, having failed "in the realm of action" re-live their careers

---

3   It could be argued that "The Aspern Papers", too, is somewhat reminiscent of Poe's "The Purloined Letter". Early in "The Aspern Papers", the narrator recalls his astonishment upon learning that Jeffrey Aspen's mistress, Juliana Bordereau, was still alive and very quietly residing in Venice, of all places. Rather than taking refuge "in an undiscoverable hole", she had "boldly settled down in a city of exhibition" (James 1908, p. 8). Like Poe's purloined letter, Juliana is hidden in plain sight.

4   As is well-known, in his preface to the New York Edition of "The Aspern Papers", James identified the germ of the story in the real-life anecdote of an American devotee of Shelley, who had gained access as a lodger into the Florentine home of Jane (known as Claire) Clairmont, the stepsister of Shelley's second wife Mary Wollstonecraft Godwin and the former mistress of Byron. Wishing to transcend his source and make the story his own, James had not only relocated the action from Florence to Venice, but had also reconfigured the national origins of both the expatriate, reclusive old lady, and the poet whose papers were the object of the lodger's obsession. In particular, by imagining Jeffrey Aspern as a compatriot, James had indulged in the seemingly daring fantasy of a possible American equivalent to a romantic bard à la Byron (James 1908, pp. xi–xii). James's specific reference to Byron might be regarded as an additional link, or allusion, to Poe who, in his early efforts as a poet, had unashamedly evoked and endeavored to emulate the author of *Don Juan*.

5   James referred with dismay to Higginson's criticism in a letter to William Dean Howells, dated 31 January 1880 (James 1999, pp. 117–19).

"through language—and [fail] again" (Veeder 1999, p. 40). I would argue that among Poe's failed, compulsive protagonists doubling as narrators, no one foreshadows the narrator of "The Aspern Papers" more closely than the self-confessed murderer in "The Tell-Tale Heart." Reading "The Tell-Tale Heart" alongside "The Aspern Papers", recognizing it as an important source for one of James's highest literary achievements, helps us gain a better understanding and a deeper appreciation of James's celebrated delineation and handling of the unreliable first-person point of view. In particular, a comparison of the two stories shows the extent to which Poe provided James with an exceptionally vivid example of a narrative voice simultaneously lucid in stating its designs and deranged in its accents and obsessions, alternatively plaintive and shockingly brutal, reticent and revealing, and how he infused and enriched it with his own profound concerns about the violation of privacy and the power imbalance between genders.

Both "The Tell-Tale Heart" and "The Aspern Papers" are narratives of despicable conduct, of maniacal singleness of purpose, resulting—directly or indirectly—in the death of a human being. Both narratives are driven forward by a note of urgency, an appeal to a hypothetic listener or reader. Try as they might to justify their actions, the two nameless men who tell their stories feel a compulsion to confess their guilt. Moreover, they are similarly obsessive in the precision and abundance of detail whereby they recount their thought processes and actions. For the narrator of "The Tell-Tale Heart", the detailed reconstruction of the methodical planning and execution of his criminal plan is the strongest argument he can wield in support of his claim to sanity. In addition, recalling the manner in which he conducted himself before and immediately after he committed murder, is the one thing that still gives him pleasure. Thinking back to that time, he cannot contain his pride and sense of accomplishment for a job well done. James's narrator, too, likes to think of himself as having been, in the early stages of his campaign to gain possession of the late Jeffrey Aspern's papers, a master planner, a consummate strategist excelling in the ancient art of war (as suggested by James's insistent use of military imagery).[6]

While denying his madness, Poe's narrator does readily admit to being nervous, indeed "very, very dreadfully" so (Poe 1990, p. 259). James's narrator, on his part, might be described as being in a state of exceptional nervous agitation, especially when he fears that the papers he has so long coveted may have been destroyed, thus eluding his grasp forever. If nervousness is a condition or a disease, whatever impairment it may cause is more than compensated, according to Poe's narrator, by his unnaturally hyperactive senses, especially his sense of hearing: "I heard all things in the heaven and in the earth" (Poe 1990, p. 260). In "The Aspern Papers", the narrator claims no such supernatural abilities, and yet, in the climax of the story, in the absence of any humanly audible sound, he does feel Juliana Bordereau's presence behind him while he is looking for the papers in her room: "It was a chance, an instinct, for *I had really heard nothing*" (James 1908, p. 118; italics mine).

Both men stalk their victims, meticulously planning their approach and carefully surveying the scene in which they will carry out their objective. In "The Tell-Tale Heart", for seven consecutive nights, the narrator practices invading the old man's most private space, his bedroom, by slowly opening the door and almost imperceptibly pushing his head in. In "The Aspern Papers", when the protagonist's confidante, Mrs. Prest, offers to take him in her gondola to look at the palace where Aspern's former mistress Juliana Bordereau and her niece Tina live (and where the papers are presumably kept), he gratefully accepts even though he has already been "to look at it half a dozen times" because "it charmed" him "to hover about the place". His subsequent, mental admission—"I had made my way to it the day after my arrival in Venice . . . laying siege to it with my own eyes while I considered my plan of campaign"—lays bare the aggressive, acquisitive nature of his

---

6 Significantly, even in defeat, James's narrator continues to think in military terms. When, towards the end of the story, Tina hints to him that she will let him have the papers if he consents to marry her, the narrator is aghast at having being outmaneuvered by a "ridiculous, pathetic, provincial old woman" (James 1908, p. 137). Tellingly, in the course of his despondent wanderings in Venice, he stops in front of, and mentally appeals to, the equestrian statue of the famous condottiere Bartolomeo Colleoni—a model of fierce masculinity—hoping to gain insight as to the best strategy to adopt in his dealings with Tina (James 1908, pp. 137–39).

gaze and qualifies his frequent visits to the place as a form of military reconnaissance (James 1908, pp. 4–5). Once he gains access into the palace by posing as a lodger, he studies its interior to get a sense of the lay of the land and takes every opportunity to get closer and closer to Juliana's quarters. Like Poe's narrator, he obsessively focuses his attention on the door that stands between him and his goal. While Juliana and her niece may have "the reputation of witches", according to Mrs. Prest (James 1908, p. 10), it is their lodger who looks like someone who "was trying to cast a spell upon" the door that leads to Juliana's part of the house, as if to will it to open (James 1908, p. 43). Thinking about Juliana and the secrets to which she holds the key, her lodger, while lingering with his candle in the hall, becomes keenly aware that his heart beats faster (James 1908, p. 44). Like a spy or a burglar planning a break-in, he gazes intently at Juliana's closed windows while sitting in the garden and pretending to read: "looking up over the top of my book". Barred from catching a glimpse of Juliana's private space, he feels that the closed windows are intended to provoke him, tantalize him. Tellingly, he imagines them as "eyes consciously closed" (James 1908, p. 44), an image reminiscent of the eye-like windows in "The Fall of the House of Usher". Later, when he learns from Tina that Juliana does indeed possess a vast amount of mementos (literary and otherwise) of Aspern, his incontrollable excitement makes itself felt through every fiber of his body: "These words caused all my pulses to throb, for I regarded them as precious evidence. I felt them too deeply to speak" (James 1908, p. 78). From that moment, he finds it increasingly difficult to contain himself. On two occasions, Tina catches his gaze as he is surveying every piece of furniture in Juliana's apartment, hoping to locate the receptacle containing the papers. The first time, he lingers in the parlor after a particularly intense interview with Juliana, even though he realizes that the exchange has taken its toll on her. Shortly afterwards, when she is taken violently ill (apparently, as a result of that interview), his sole, all-conquering thought is that he has a chance to be in even closer proximity to his prize: "I didn't waste my time in considering Juliana . . . I turned my eyes once more all over the room, rummaging with them the closets, the chests of drawers, the tables". Even when he becomes embarrassingly aware that the movements of his eyes have been followed closely by Tina ("Miss Tina at once noted their direction and read, I think, what was in them"), he cannot stop (James 1908, pp. 104–5).

Unlike James, Poe provides us with the bare minimum of information about the two actors in his drama. Other than the fact that they live in the same house and that there seems to be a significant difference in age between them (the narrator primarily identifies his victim by his old age), we know precious little. As Christopher Benfey pointed out (Benfey 1993, p. 32), we never learn anything about the nature of the relationship between the old man and the narrator: are they master and servant? Are they father and son or grandfather and grandson? Could they be close friends living together, or even lovers? In addition, we do not know what nationality they are (by contrast, national identity, or lack thereof as the result of a long residence abroad, figures quite prominently in "The Aspern Papers"). Nor do we know in what country the story unfolds. A reference to the old man's fear of robbers, the quick arrival of policemen on the scene of the crime at the end, and the narrator's lie about the old man being "absent in the country" appear to indicate an urban setting (Poe 1990, p. 262). In stark contrast to other tales by Poe ("The Masque of the Red Death" comes to mind), even the interior of the house in "The Tell-Tale Heart" remains conspicuously shadowy and barren except for a few items related to the execution and covering up of the crime (a latch, a door, a lantern, a mattress, a tub, the floorboards, a chair, etc.). It is almost as if the narrator's exclusive focus on the old man and his baleful eye had banished nearly everything else from his ken.

If Poe renders his setting as rarefied as possible (to the point of turning it almost into a non-place or any place), James lavishes his evocative powers on Venice, the backdrop to his story. He does so by conjuring up the city's rich literary associations (most notably

Shakespeare and Byron, and possibly Poe's own Venetian tale "The Assignation")[7] and history, as well as by making the most of his own first-hand knowledge of the place. And while he draws from the Gothic tradition (including Poe's "The Fall of the House of Usher") to depict the interior of the Bordereau palace, he simultaneously portrays it in realistic detail as a neglected, run-down, dusty building, which is fairly representative of the decaying, impoverished, late-nineteenth-century Venice.

Where James does closely follow the Poe model is in having his protagonist conspicuously withhold information about himself. As in the case of "The Tell-Tale Heart", in "The Aspern Papers", the narrator never reveals his name (either his real name or the false name with which he introduces himself to the Misses Bordereau), nor his age or place of birth or residence. Like his counterpart in Poe, he identifies himself almost exclusively through his commitment to his purpose. What he clearly regards as relevant about himself—and hence, worthy of being shared—is his devotion to Jeffrey Aspern and his determination to get hold of whatever papers the late poet may have left behind. Similarly, in "The Tell-Tale Heart", the narrator is keen on sharing with us his obsession with the old man's "evil eye", an obsession he fears many would consider pathological. James's narrator, on his part, is no less concerned about the possibility that his conduct might be judged as being beyond the pale.

Both men have recourse to dissimulation in order to achieve their ends. Like actors, they need to play a part convincingly. Specifically, they need to convince the targets of their designs that they do not pose a threat to their well-being. Thinking back to the days preceding the murder of the old man, Poe's narrator wishes there had been an audience for his consummate performance: "You should have seen . . . with what dissimulation I went to work! I was never kinder to the old man than during the whole week before I killed him" (Poe 1990, p. 260). That hypothetical audience could not have failed to be impressed by the act the narrator put on every morning when he went into the old man's room: "[I] spoke courageously to him, calling him by name in a hearty tone, and inquiring how he had passed the night" (Poe 1990, p. 260). What better proof of his success than the fact that the old man apparently suspected nothing of his real intentions?

Similarly, James's narrator is keenly aware that the success of his plan is contingent upon his ability to hide his real intentions from Juliana. As he confides to Mrs. Prest (his audience of one), he believes he can "arrive at [his] spoils only by putting [Juliana] off her guard" by means of "ingratiating diplomatic arts". As he unashamedly puts it, "[h]ypocrisy, duplicity are my only chance". Armed with an assumed name (a *nom de guerre* "neatly engraved" on a visiting-card), he does indeed gain access into the seemingly impregnable fortress he had studied so closely (James 1908, pp. 11–12). Cunning and unencumbered by scruples, he is an American Ulysses as well as his own Trojan horse.

In both "The Tell-Tale Heart" and "The Aspern Papers", the obstacle to what the protagonists believe would be their ultimate fulfillment is represented by a person of advanced age with whom they share living quarters. For Poe's narrator, the obstacle is a nameless old man, or rather the man's veiled eye, which he describes as vulture-like and evil. In James's story, Juliana is the sole custodian of private writings (presumably love letters) penned by Jeffrey Aspern, the American poet the narrator literally worships. As a self-professed devotee and high priest of the Aspern cult, the narrator believes he would be a worthier guardian and executor of Aspern's literary remains. Even though he never contemplates doing Juliana any bodily harm—should she refuse to part with the papers—the military metaphors that come so easily to his mind betray a barely-suppressed hostility towards her, as do the many instances in which he foreshadows—and perhaps secretly wishes for—her death.[8]

---

7　On the possible sources for the representation of Venice in "The Aspern Papers", see, for example, Rosella Mamoli Zorzi's "The Aspern Papers: From Florence to an Intertextual City, Venice" (Mamoli Zorzi 2014, pp. 103–11).

8　Referring to the narrator's last, fatal encounter with Juliana, Joseph Church has stated that the narrator's action "amounts to a kind of unconscious murder-wish" (Church 1990, p. 33).

In the presence of persons advanced in years, the protagonists of "The Tell-Tale Heart" and "The Aspern Papers" become fixated on physical signs of infirmity and decay. Whether evil or not, the old man's "pale blue eye, with a film over it" in "The Tell-Tale Heart" (Poe 1990, p. 260) is inescapably a marker of bodily degeneration, the visible manifestation of the weakening or loss of function of an essential organ. It offers incontrovertible evidence of the aging process and prefigures its inevitable outcome: death. For these reasons, it is a sight that infuriates, disgusts, and terrifies the narrator. He cannot bear to look at it because it is like staring death in the face (his own as much as the old man's). Similarly, it is the face of death (given the age-old portrayal of death as a human skull), that the narrator of "The Aspern Papers" imagines staring at him from behind the green veil that covers most of Juliana's visage. That veil, the narrator recalls, "created a presumption of some ghastly death's-head lurking behind it. The divine Juliana as a grinning skull" (James 1908, pp. 23–24). While the search for the possible literary source of Juliana's veil has understandably focused on Hawthorne's "The Minister's Black Veil" (Hawthorne 1958) and *The Blithedale Romance* (Hawthorne 1983), I would argue that the film covering the old man's eye in "The Tell-Tale Heart" is also a strong contender. Juliana's "horrible green shade" (James 1908, p. 23) practically echoes the old man's "hideous veil" through which only the color blue, however faintly, can be perceived (Poe 1990, p. 261). Although in "The Tell-Tale Heart", common sense—in the absence of medical opinion—would suggest that the veil (possibly a cataract) covering the old man's eye significantly impairs his vision, the narrator seems to believe (and fear) that the veiled eye is actually all-seeing and that nothing can escape its surveillance. If anything, from the narrator's point of view, the veil renders the old man's gaze an even more insidious threat, because he cannot penetrate that barrier and follow the eye's movements, while the old man can apparently look deep into the narrator's eyes and perhaps read his innermost secrets. In "The Aspern Papers", the narrator's first impression, as he is confronted with Juliana's veiled face, is that "she had put it on expressly, so that from underneath it she might take me all in without my getting at herself" (James 1908, p. 23). Baffled in his effort to read her face as one would a text, he confesses that "the old woman remained impenetrable and her attitude worried me by suggesting that she had a fuller vision of me than I had of her" (James 1908, p. 27). As Gero Bauer has pointed out, "considering the importance of the gaze in gendered power relations, it is crucial to observe that the editor is repeatedly deprived of his right to look" (Bauer 2016, p. 168). Like her face, Juliana's vision is shrouded in mystery. When the narrator, upon learning that Juliana has been taken gravely ill after talking to him, tells Tina that it would be better if her aunt were "spared the sight" of him, Tina replies indignantly: "'The sight of you? Do you think she can *see*?'" (James 1908, p. 104). And yet, in the climax of the story, when Juliana surprises the narrator in her apartments searching for the papers, she fixes her angry eyes—visible for the first time—on him and recognizes him instantly. It is of course possible that she assumes it is him, given the circumstances, but we can never know for sure.[9] Even though Tina's protestation seems to suggest that Juliana is blind or nearly blind, when the narrator finds himself transfixed by her stare, he does not doubt for a moment that she can see him. Moreover, on previous occasions, he had felt, with no small discomfort, that she was looking at him and assessing him. Significantly, it is Juliana herself who has encouraged that impression. When the narrator, a few months into his residence at the palace, sends his servant to inquire if he can pay a visit to the ladies, the surprising report he receives is that Juliana "had been moved out into the sala and was sitting by the window that overlooked the garden" (James 1908, p. 87). The garden, which the narrator has revived at a considerable expense as a way of ingratiating himself to his landlady and her niece, is where he has spent considerable time, thus potentially exposing himself to her scrutiny. Shortly afterwards, when Juliana tries unashamedly to pressure her lodger into paying his exorbitant rent for another six months, she assures him

---

9    Another possibility is that Juliana is pretending that she can see. Too proud to admit she has lost her eyesight, she might be mimicking the behavior of a person who can see, much like Flora Saunt in the final scene of "Glasses" (James 1996).

she is not exaggerating the value of her property. She has made sure of it, she emphasizes, because she has personally re-examined it with *her own eyes*: "'This house is very fine; the proportions are magnificent. To-day I wanted to look at this part again. I made them bring me out here. . . . I wanted to judge what I'm letting you have'" (James 1908, pp. 90–91). Later, after trying to sell him a small portrait of Jeffrey Aspern for a preposterous sum of money, she concludes their interview by reiterating her intention to follow his every move whenever possible: "'To-morrow I shall come out again. I want to be where I can see this clever gentleman. . . . I want to watch you—I want to watch you!'" (James 1908, p. 98).

At the outset of "The Tell-Tell Heart", the murderer feels the need to remove from his actions even the slightest suspicion of a mercenary motive. For the old man's gold, he claims, he "had no desire". Later, when three police officers arrive on the scene, he makes it a point of showing them the old man's "treasures, secure, undisturbed" (Poe 1990, pp. 260, 262). The possible taint of greed, it seems, would be almost as unbearable as the taint of madness. His motives had nothing to do with something as debased and degrading as money. Quite the contrary: no motive could have been more elevated and heroic than what inspired him: he was battling evil. We never learn how wealthy the old man actually was, but the word "treasures" suggests that, at least from the murderer's point of view, his fortune may have been considerable. The old man himself appears to have deemed his fortune, whatever its size, worth protecting, given that the shutters in his room "were close fastened, through fear of robbers" (Poe 1990, p. 260). By contrasting his own disregard for "gold" with the old man's concern about his possessions, the narrator implicitly asserts his spiritual superiority over his victim.

Unlike Poe's old man, Juliana is in very precarious financial straits. So much so that the narrator's confidante, Mrs. Prest, had interested herself in her (and Tina's) predicament. From the perspective of Mrs. Prest and her circle, the Misses Bordereau are a charity case. When the narrator expresses his interest in taking rooms in her palace, Juliana immediately senses an inordinate eagerness in his tone and charges him an exorbitant price. Whatever might be wrong with her eyesight, her hearing seems wonderfully acute (perhaps nearly as acute as Poe's narrator fancies his to be), as she herself points out when she rebukes her interlocutor for speaking too loud: "'I hear very well'" (James 1908, p. 25). Although keenly aware that he will be paying a grossly inflated sum for his lodgings, the narrator accepts Juliana's terms and clearly likes to portray himself as indifferent to money. Although he repeatedly emphasizes (both in his internal musings and while talking to Juliana and Tina) that he is not wealthy, he seems determined to demonstrate that parting with money gives him no pain because it is in the service of the better good. However preposterous, the rent he will pay is an investment that will place him within reach of his goal, as well as a tribute to Jeffrey Aspern: "if she had asked five times as much I should have risen to the occasion, so odious would it have seemed to me to stand chaffering with Aspern's Juliana" (James 1908, p. 28). Like his counterpart in "The Tell-Tale Heart", the narrator in "The Aspern Papers" believes that his indifference to money elevates his motives, rendering them purer and nobler. It brings him closer to his literary God, Aspern. Significantly, he contrasts his lack of concern about pecuniary matters with Juliana's seemingly miser-like determination to extract from him as much money as she possibly can. Although he does take some responsibility, in as much as he put the idea into her head that he would take the rooms at any cost, he still suggests that her character (clearly not as strong as his) made her susceptible to the corruptive influence of money. And while he claims to be shocked and disappointed by her greed, it is not inconceivable that her attitude should secretly please him, for it confirms him in his deep-seated belief that he would be a much worthier custodian of the papers than she is. It seems no accident that his account should include several scenes in which Juliana appears to be exclusively and almost obscenely focused on financial gain. By recreating her speech on those occasions, he gives the impression that money (preferably in the form of shining gold currency) is constantly on her mind and on her lips. From their very first encounter, she utters words that in their bold acquisitiveness

ill accord with the image of a poet's muse: "'You may have as many rooms as you like—if you will pay me a good deal of money'" (James 1908, p. 28).

Moments later, when Tina joins her, Juliana is practically giddy at the thought of the bargain she has just completed, and seems to savor the very sound of the number that spells out the successful outcome of the transaction: "'He'll give three thousand—three thousand to-morrow!'" (James 1908, p. 29). The narrator, on the other hand, presents himself as the dedicated devotee of art for whom money is almost too vulgar to be mentioned. Here and elsewhere, he is oblivious to his own rapacity, as well as to the note of violence that colors his desire to supplant Juliana as the owner of Aspern's literary relics: "I would pay her with a smiling face what she asked, but in that case I would make it up by getting hold of my 'spoils' for nothing" (James 1908, p. 28). Tellingly, his stance of moral superiority does not waver even after learning that Juliana is so keen on accumulating as much money as she can so that Tina will be provided for after her death. Instead of sensing, and empathizing with, the note of despair with which Juliana, later in the story, asks him if he will extend his stay, he debases her motives: "'Have you come to tell me that you'll take the rooms for six months more?' she asked as I approached her, startling me by something coarse in her cupidity" (James 1908, pp. 87–88). While he is convinced that his own conduct is justified and ennobled by the purpose it serves (preserving Aspern's papers for posterity), he completely fails to acknowledge that Juliana's demands, however crude, stem from her love for another human being. Nor does he understand that Juliana, too, is making a sacrifice: in order to secure Tina's future, she is willing to appear as greedy as Scrooge. One does not need to subscribe to the theory that Tina is Juliana and Aspern's illegitimate daughter[10] in order to agree with John Carlos Rowe when he observes that "Juliana Bordereau has the natural instincts of a mother to provide for her child" (Rowe 1984, p. 109).

As more than one critic has noted, an undercurrent of sexual hostility transpires from the narrator's attitude towards women, especially those he sees as rivals for the love of Aspern. Early on, he pointedly elevates his devotion for Aspern well above the frenzied, carnal passion of the women who had pursued the poet during his short adult life: "Half the women of his time, to speak liberally, had flung themselves at his head . . . 'Orpheus and the Mænads!' had been of course my foreseen judgment when first I turned over his correspondence. Almost all the Mænads were unreasonable, and many of them unbearable" (James 1908, p. 7). Later, he intimates that Juliana might have violated the rigid boundaries of what the society of her time regarded as acceptable behavior for a young lady, while implicitly absolving Aspern of any guilt in that transgression: "There hovered about her name a perfume of impenitent passion, an intimation that she had not been exactly as the respectable young person in general" (James 1908, p. 48). If not kept firmly under control, sexuality, he seems to imply, obfuscates the mind and thus prevents a genuine appreciation of Aspern's poetry. For this reason, he tries to distinguish his own devotion of Aspern from that of all women by moving it to a higher plane, as far removed as possible from the sphere of the body and physical desire, hence the religious terminology to which he has recourse when referring to the American poet. Thus, Aspern becomes a god and the narrator, the self-appointed minister of its cult. And yet on more than one occasion, the words with which he expresses his supposedly quasi-religious fervor betray a different (and from his perspective, lower) type of attraction. While he never quotes a single line of poetry penned by his idol (surprisingly, for a literary historian/biographer), he clearly takes pleasure in mentioning that, in addition to being "one of the most genial men" of his time, Aspern was also "one of the handsomest" (James 1908, p. 6). When Juliana shows him the portrait of Aspern, it is he, not her, who loses control of his body when confronted with a reproduction of the man's beautiful face at the apparent age of 25: "I possessed myself of it with fingers of which I could only hope that they didn't betray the intensity of their clutch . . . At the first glance I recognized Jeffrey Aspern, and I was well aware

---

10    This interpretation was first presented by James Gargano in "'The Aspern Papers': The Untold Story" in 1973 (Gargano 1973).

that *I flushed* with the act" (James 1908, pp. 93–94; italics mine). Jealous of the intimacy that Juliana likely experienced with the handsome man of the portrait, the narrator can only find satisfaction by appropriating the only extant record of that intimacy, namely Aspern's letters to her. Symbolically, he wants to substitute himself for her and enjoy, albeit vicariously, the supposedly physical contact with Aspern inscribed in those papers. This involves, inevitably, violating her space (her own private apartment where the papers are hidden) and, indirectly, her person, as if to emphasize her unworthiness as the recipient of Aspern's love.

Like the murderer in "The Tell-Tale Heart", James's narrator ultimately takes pleasure in asserting his power over his victim (as Juliana may be justifiably described). Breaching her privacy and attempting to steal the papers might even be regarded as "a figurative sexual assault" (Miller 2005, p. 21). While there is no overt sexual connotation to his actions, the murderer in Poe's story clearly derives enormous gratification from his nightly forays into the old man's room, the man who, by his own admission, he "loved" (Poe 1990, p. 260). He certainly enjoys recounting how he did it, in a moment-by-moment reconstruction that smacks of exhibitionism. What particularly fills him with pride and pleasure, is to think back to the way he slowly but surely pushed his head into the old man's room, how he literally penetrated the old man's most intimate domain: "and then I thrust in my head. Oh, you would have laughed to see how cunningly I thrust it in!" (Poe 1990, p. 260). On the fateful eighth night, he is almost heady with excitement ("I could scarcely contain my feelings of triumph"), which is probably why his thumb slips upon the tin fastening of the dark lantern he is carrying, making a noise, just after he had managed to push his head into the room (Poe 1990, p. 260). But when the old man awakes and cries "'Who's there?'", the murderer does not retreat. Sensing his victim's terror, he feels even more empowered and thrilled. Even though this is never specified in the story, it is possible that the victim, in addition to being old, might be bed-ridden, and as such, supremely vulnerable. Interestingly, in his only reference to any interaction with the old man during the day, the narrator mentions that every morning he went into his room. The only time he alludes to the old man as being anywhere else is when he is questioned by the police, but what he tells them (that the old man was in the country) is clearly a fabrication. What is certain is that the murderer enjoys the old man's helplessness enormously and sadistically. And since, as he readily admits, he too has often been preyed upon by those very nightly terrors, his mastery over the old man gives him the additional pleasure (and the illusion, as it turns out) of having conquered his own demons. Terrified of death, he feels he has *become* death or, at the very least, its agent. Imagining the old man's attempts to find reassuring explanations for the noise that had awakened him, the murderer notes with obvious satisfaction that it had all been "*All in vain*; because Death, in approaching him, had stalked with his black shadow before him, and enveloped the victim. And it was the mournful influence of the unperceived shadow that caused him to feel—although he neither saw nor heard—to *feel* the presence of my head within the room" (Poe 1990, p. 261).

Similarly, in "The Aspern Papers", the narrator believes that the old person with whom he is dealing is being stalked by death. Upon encountering Juliana for the first time, and taking in the impression of physical decay so advanced that it can hardly be measured in human years, it occurs to him that "death might take her at any moment" (James 1908, p. 24). In retrospect, that passing thought acquires an ironic connotation, because, for all intents and purposes, he himself turns out to be, quite literally, the death of her. Like Poe's murderer, he does, however unintentionally, death's job. At first worried that Juliana's seemingly imminent death might put Aspern's papers out of his reach, he promptly reassures himself that her demise would actually facilitate the transfer of those papers into his possession: "She would die next week, she would die to-morrow—then I could pounce on her possessions and ransack her drawers" (James 1908, p. 24).

In "The Tell-Tale Heart", the narrator finds himself unable to carry out his plan for seven consecutive nights because an eyelid conceals the object of his hatred and terror, namely the old man's "evil" eye. It is only when the thin ray of light issuing from his

semi-closed lantern finds that eye wide open that the narrator is goaded into action. It is almost as if that eye had finally accepted the challenge posed by the narrator's light, thus giving him license to launch his attack. In the climactic episode of "The Aspern Papers", the narrator returns to the palace late at night after taking a walk, hoping to find Tina waiting for him with news about her aunt's health. But she is nowhere to be seen and he is considerably disappointed when, upon crossing the sala, his lamp finds "nothing satisfactory" to show him (James 1908, p. 115). However, after entering Juliana's apartment, he perceives that the door of her room is open and this sight propels him forward. Like Poe's narrator, he slowly and cautiously enters the room. He too relies on his lamp, as if it were an extension of his eyes (and a projection of his desire), letting "the light play on the different objects as if it could tell me something" (James 1908, p. 116). He too describes his actions as the inevitable response to a provocation: the door left open as if to invite him in; the secretary possibly left unlocked by Tina as if to dare him to test its mechanism and see if it might open and disclose its secrets. What confronts him, instead, when he looks over his shoulder, is the accusatory and appalled stare of Juliana, who has miraculously risen from her sickbed, like an aged version of Poe's female revenants. In a specular reversal of the "Tell-Tale Heart" climax, the human glare (Juliana's) triumphs over the intruder's device: "I almost let my luminary drop and certainly I stepped back". And it is the intruder who is petrified with fear upon finding himself under the scrutiny of Juliana's eyes, startlingly visible for the first and last time. Even more eloquently than her words ("Ah you publishing scoundrel!"), her eyes, wide open, confront the narrator with the enormity of his misconduct. Reading much more deeply into his soul than he ever did himself, Juliana falls back into Tina's arms when he approaches, "with a quick spasm, as if death had descended on her", as indeed proves to be the case shortly afterwards (James 1908, p. 118).

As William Veeder has observed, "The end in 'The Aspern Papers' reveals dramatically what many other James tales indicate—how much his failed protagonists resemble those of an American precursor whose influence on James has yet to be appreciated fully: Edgar Allan Poe" (Veeder 1999, p. 40). In Poe, especially in "The Tell-Tale Heart", James found the prototype of an obsessive narrator, oscillating between exhibitionism and shame, maniacal reporting and self-deception. Most of all, he found the model for a protagonist who is driven by an irresistible urge to escape from isolation by establishing meaningful contact with his fellow human beings through the medium of a confession. And yet, great as the desire to unburden themselves is in both narrators, it coexists with an irrepressible need to justify, or at least to try to justify, their actions, which ultimately makes their narratives all the more human and compelling. However, in reinventing the Poe narrator, James considerably toned down the frantic, oral quality, the theatricality of the original (which had made it ideal for reading aloud), endowing it with a more controlled, subdued tone, appropriate for a man seemingly incapable of strong feelings for any living creature (unlike Poe's narrator who, in his own words, "loved" the man he killed) and who suffers a crushing defeat, namely the loss of the papers. That loss, for the narrator, is the closest thing to the loss of a loved one, a feeling the likes of which (in terms of intensity and depth), he has probably never experienced.[11] Until the very end of his narrative, as he looks at the portrait of Aspern hanging over his writing desk, he remains oblivious to another, much more glaring absence in his life—love for another human being—for which even the possession of the papers would have been a poor surrogate.

**Funding:** This research received no external funding.

**Institutional Review Board Statement:** Not applicable.

---

[11] Interestingly, when James's narrator considers that he might bring himself to marry Miss Tina (a possibility that had initially horrified him) as a way to secure the papers, he sounds not unlike those male characters in Poe's fiction who seem to conceive of marriage as an unavoidable compromise, based on anything *but* physical attraction or passion. A typical example is the protagonist/narrator of "The Black Cat" who has recourse to the negative form to describe the nature of his relationship with his wife: "I married early, and was happy to find in my wife a disposition not uncongenial with my own" (Poe 1990, p. 254).

**Informed Consent Statement:** Not applicable.

**Conflicts of Interest:** The author declares no conflict of interest.

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
