# Peer review of "Echoes of the Heart: Henry James’s Evocation of Edgar Allan Poe in “The Aspern Papers”"

_humanities, doi:10.3390/h10010055_

Round 1

Reviewer 1 Report

This is a clearly written and well organized essay. The author convincingly argues that Poe’s story influenced the composition of James’s “The Aspern Papers.” The essay is grounded in a substantial body of scholarship to which it makes a valuable contribution. I can find no fault with it.

Author Response

Thank you for your remarks.

Reviewer 2 Report

I think the author makes a strong case for revisiting James and Poe. In particular the author calls our attention to an under-discussed context for thinking about "The Aspern Papers." The essay is will argued, written, and researched. While I believe the comparison between James's story and Poe's is persuasive and apt, I would like the author to push the argument further a bit. This could mean a sentence or paragraph more. In particular, my question is thus: what do we gain from reading James's AP in relation to Poe's TTH? What are the ultimate fruits of the comparison the author offers? The author is right to point out that the connect between each story has not been made, but the author does not give a reason why it should be. Do we learn something about James's decadent imagination? Something about the shape of his career? Something about his relation to Poe or American literature? Maybe something about his relation to the gothic or ghost story? Or maybe it is something stylistic or about James's artistic sensibility? It would be useful if the author anticipated the stakes of the paper in the beginning, before summarizing the scholarship, and then return to it at the end. Could be several sentences, a paragraph, or a couple paragraphs. But I would like the author to simply further justify the value of their comparative reading. 

Small point, they call Leland Person "Leland" and "Person" at times on the second page. 

Author Response

Thank you for your remarks.

Reviewer 3 Report

This essay examines a surprisingly understudied source for Henry James's canonical novel, The Aspern Papers. In short, a remarkably astute and compelling connection is made between Poe's famous short-story "The Tell-Tale Heart" and James's novel in this essay. The author's thesis is clearly established by recovering biographical and narrative ties that link James to his forebear which, without question, prove that Poe was an inspiration for James as he penned Aspern. Moreover, this essay is well-written, and there are only a small handful of minor typos that will need to be corrected before publication.

I have one somewhat minor suggestion, in addition to a broader provocation (which the author does not necessarily need to engage). In terms of the first recommendation, on record, in the New York Edition, James credits Byron first and foremost as the key for better understanding Aspern. It may be useful to establish or clarify this ostensible inspiration for James's novel a bit earlier to make your important scholarly intervention (e.g., the connection to Poe) clearer a bit closer to the beginning of the essay.

In this spirit, though I'd hesitate to require this suggestion, especially if there are spatial constraints, offering a more comprehensive account of hidden/potential sources for Aspern might help clarify how distinctive this essay is in comparison to older/extant scholarship.

My comments that follow do not need to be addressed, but may help establish the scholarly literature this essay is seemingly engaged with. The author may consider what is seemingly a burgeoning subfield of literary history that has convincingly outlined where and how Alexander Pushkin's "Queen of Spades" (Pushkin's initials are ASP, to offer just one of numerous conspicuous ties to Aspern) is also an overlooked source for James's novel. None of this scholarship has considered how race may have prompted what James "hid," but this is certainly not incidental--from a biographical perspective.

Juliana may be modeled after Natalia Pushkina, Pushkin's daughter, who sold her father's letters to Turgenev (one of James's dear friends and interlocutors) in this era. Pushkin's black African ancestry may be tied to the "secrets" haunting both the "Queen of Spades" (clearly signaled in the title), and even more conspicuously in Aspern

A second reference pertaining to race that the author might consider exploring is James's cousin, Clarence King, a white man who passed as black to marry an African American woman around the time Aspern was published. Martha Sandweiss's Passing Strange, a biography of King's decision to live as a black man, mentions James throughout the narrative. King was a part of a social group James was intimately connected with from a young age. The nickname they gave themselves?: "The Queen of Hearts."

Even without the conspicuous "heart" interconnection, none of this material is divorced from how Poe might be understood. In Toni Morrison’s classic, Playing in the Dark (1992), she argued, “No early American writer is more important to the concept of American Africanism than Poe.”

For sources that have tied James and Pushkin together, the author might consider citing and/or reading the following sources:

  1. A.D. Briggs, “Alexander Pushkin: A Possible Influence on Henry James,” Forum for Modem Language Studies, Volume VIII, Issue 1 (January l972), 52 – 60.

  1. Andrew R. Durkin, “Pushkin among the Edwardians: Revision and Renewal of Cultural Memory in James and Conrad” in Hendrik von Gorp and Ulla Musarra-Schroeder, eds., Genres as Repositories of Cultural Memory (Amsterdam: Rodopi, 2000), 67 – 76.

  1. Hana Voisine-Jechova, “Le secret de la vieille dame: A. S. Puškin et Henry James,” Revue de Littérature Comparée69.3 (1995), 263 – 71.

  1. Neil Cornwell, “Pushkin and Henry James” in The Literary Fantastic: From Gothic to Postmodernism (New York: Harvester Wheatsheaf, 1990), 113 – 39.

  1. Jean Norris Scales, “The Ironic Smile: Pushkin’s ‘The Queen of Spades’ and James’ ‘The Aspern Papers,’” CLA Journal 34.4 (June, 1991), 486 – 490.

  1. Neil Cornwell, “Pushkin and Henry James: Secrets, Papers and Figures (The Queen Of Spades, The Aspern Papers, and The Figure in the Carpet)” in Joe Andrew and Robert Reid, eds., Two Hundred Years of Pushkin, Volume 3 (Amsterdam: Rodopi, 2004), 193 – 208.

  1. Joseph S. O’Leary, “Anathemata for Henry James,” English Literature and Language 36 (1999), 63 – 99.

  1. Andrew R. Durkin, “Henry’s James’s Response to Pushkin: ‘Pikovaia Dama’ and ‘The Aspern Papers’” in Robert A. Maguire et al., eds., American Contributions to the XII International Congress of Slavists (Bloomington: Slavica, 1998), 52 – 61.

  1. John Bayley’s Chapter 3: “Poems of the South” in Pushkin: A Comparative Commentary (Cambridge: Cambridge University Press, 1971), 84 – 85.

  1. Philip Ross Bollock, “Untranslated and Untranslatable? Pushkin’s Poetry in English, 1892 – 1931,” Translation and Literature 20 (2011), 348 – 72.

For sources on Poe and race, see, for example:

Dan Shen, “Edgar Allan Poe’s Aesthetic Theory, the Insanity Debate, and the Ethically Oriented Dynamics of ‘The Tell- Tale Heart,’” Nineteenth-Century Literature 63, no. 3 (2008): 321–45.

Aaron Matthew Percich, “Irish Mouths and English Tea-pots: Orality and Unreason in “The System of Doctor Tarr and Professor Fether,” Poe Studies Vol. 47 (2014): 76-99

Brian Yothers, “Arabs, Arabesques, and America: The Place of Poe in Studies of Literary Orientalism,” Poe Studies, Vol. 47 (2014): 115-119

Courtney Novosat, “Outside Dupin’s Closet of Reason: (Homo)sexual Repression and Racialized Terror in Poe’s ‘The Murders in the Rue Morgue,’ Poe Studies Vol. 45, No. 1 (2012): 78-106

  1. Gerald Kennedy, “‘A Mania for Composition’: Poe’s Annus Mirabilis and the Violence of Nation-Building,” American Literary History Vol. 17, No. 1 (Spring, 2005): 1-35.

Joan Dayan, “Poe, Persons, and Property,” American Literary History Vol. 11, No. 3 (Autumn, 1999), 405-425.

I look forward to seeing this essay on Poe and James in print, it was a pleasure to read.

Author Response

Thank you for your remarks.